# Differences between Kazak Cheeses Fermented by Single and Mixed Strains Using Untargeted Metabolomics

**DOI:** 10.3390/foods11070966

**Published:** 2022-03-26

**Authors:** Yandie Li, Jianghan Wang, Tong Wang, Zhuoxia Lv, Linting Liu, Yuping Wang, Xu Li, Zhexin Fan, Baokun Li

**Affiliations:** 1School of Food Science and Technology/Key Laboratory of Xinjiang Phytomedicine Resource and Utilization of Ministry of Education, Shihezi University, Shihezi 832000, China; 20192011019@stu.shzu.edu.cn (Y.L.); wangjianghan@stu.shzu.edu.cn (J.W.); 20202011021@stu.shzu.edu.cn (T.W.); lvzhuoxia@stu.shzu.edu.cn (Z.L.); liulinting@stu.shzu.edu.cn (L.L.); wangyuping@stu.shzu.edu.cn (Y.W.); lixu@stu.shzu.edu.cn (X.L.); zhexin_fan@shzu.edu.cn (Z.F.); 2Guangdong Yikewei Biotech Co., Ltd., Guangzhou 510520, China

**Keywords:** *Lacticaseibacillus paracasei*, *Kluyveromyces marxianus*, flavor compounds, ripening, metabolic mechanism, metabolomics

## Abstract

Mixed fermentation improves the flavor quality of food. Untargeted metabolomics were used to evaluate the impact of mixed fermentation and single-strain fermentation on the volatile and non-volatile compound profiles of Kazak cheese. *Lacticaseibacillus paracasei* SMN-LBK and *Kluyveromyces marxianus* SMN-S7-LBK were used to make mixed-fermentation cheese (M), while *L. paracasei* SMN-LBK was applied in single-strain-fermentation cheese (S). A higher abundances of acids, alcohols, and esters were produced via mixed fermentation. Furthermore, 397 differentially expressed non-volatile metabolites were identified between S and M during ripening. The flavor compounds in mixed-fermentation cheese mainly resulted from ester production (ethyl butanoate, ethyl acetate, ethyl octanoate, and ethyl hexanoate) and amino acid biosynthesis (Asp, Glu, Gln, and Phe). The metabolites were differentially expressed in nitrogen metabolism, D-glutamine and D-glutamate metabolism, phenylalanine metabolism, D-alanine metabolism, and other metabolic pathways. The amount of flavor compounds was increased in M, indicating that *L. paracasei* SMN- LBK and *K. marxianus* SMN-S7-LBK had synergistic effects in the formation of flavor compounds. This study comprehensively demonstrated the difference in metabolites between mixed-fermentation and single-strain-fermentation cheese and provided a basis for the production of Kazak cheese with diverse flavor characteristics.

## 1. Introduction

Kazak cheese is a traditional artisanal fermented cheese produced by Kazak minority farmers on a small scale in Xinjiang, China, using traditional practices and raw bovine or goat milk. Fresh milk with (or without) natural starter (yogurt from a previous fermentation) is fermented and curdled into yogurt in sheepskin bags. After removing whey, dehydrating, and shaping, the curd undergoes ripening of 30–60 d to make cheese [1]. Autochthonous microflora in both starters and sheepskin fermentation vessels dominate the processes of fermentation and ripening of typical Kazak cheese [2]. Endogenic microbial starters, mainly lactic acid bacteria (LAB), play a vital role in the fermentation process, contributing to the quality and physiological activity of cheese [3]. Additionally, an increasing number of studies have evidenced that non-starter LAB (NSLAB), yeast, and mold are cheese-inhabiting microbiota that are crucial in forming the typical, markedly acidic flavor of this cheese [4]. Investigations on the diversification of cheese flavor suggest that mixed fermentation by multiple strains might be an alternative way to improve flavor properties.

In a Cremoso cheeses model, the effect of *Lacticaseibacillus paracasei* on flavor was assessed using a single culture [5]. Furthermore, products fermented by *L. paracasei* have been reported to have potential prebiotic effects, such as the potential to colonize the gut [6], antioxidant activity [7], and antimicrobial activity [8]. Yeasts are widely dispersed in the cheese production environment and processes and considered to be adjunct starter cultures contributing to aroma development through the proteolysis process [9,10,11]. *Lacticaseibacillus* and *Kluyveromyces* are the major flavor contributors in some regional Kazak cheeses [12]. The comparison of mixed fermentation and single-strain fermentation will be helpful for attempts to develop diverse flavor characteristics in dairy products [13].

Flavor characteristics are an important component of cheese quality. Cheese flavor during ripening is mainly affected by glycolysis, proteolysis, and lipolysis [14,15]. Among these, protein hydrolysates, such as small molecular peptides and free amino acids, can affect the taste of cheese, and amino acids are flavor precursors that can be converted into typical volatile compounds [16]. Cheese flavor components are mainly composed of volatile and nonvolatile compounds. Volatile (alcohols, acids, esters, aldehydes, and ketones) and nonvolatile (organic acids, amino acids, reducing sugars, nucleotides, and polypeptides) flavor compounds contribute to the aroma and taste of cheese, respectively. Therefore, obtaining a comprehensive understanding of the contributions of different metabolite classes through qualitative and quantitative analyses of metabolites is necessary.

The mixed fermentation of LAB and yeast is an effective means to improve the flavor quality of fermented food. LAB and yeast are often used in combination as starter cultures for functional fermented foods [17]. The mixed fermentation system of LAB and yeast is used to improve the aroma intensity [18], increase the nutritional value [19], and produce functional components [20]. Sensory characteristics deeply depend on the composition and content of aroma compounds as well as nonvolatile secondary metabolites produced in the fermentation process [21]. The analysis of characteristic flavor compounds in mixed fermentation is helpful to understand the fermentation performance of mixed fermentation and its influence on cheese metabolic profile, which is conducive to the directional control of the production of characteristic cheese. Cheese metabolite profiles are used to show differences in metabolic pathways relating to the production and effect of key aroma compounds [22], which need to be systematically investigated. Metabolomics produces a huge amount of raw data from different high-precision instruments, and extracting effective information from these complex data is difficult [23]. Therefore, chemometric techniques are needed in order to interpret and classify sample sets, with principal component analysis (PCA) and partial least squares (PLS) being the most commonly used examples [24]. Metabolomics, which can identify and quantify small-molecule substances through technology such as mass spectrometry (MS), has been widely used in the evaluation and analysis of foods fermented with LAB and the fermentation performance of these strains [25]. Metabolomics research on fermented cheese production [26,27,28] can identify flavor formation and associate it with metabolites through multivariate analysis; thus, it is useful in the design of high-quality foods [29].

Although the LAB and yeast mixed fermentation system has been successfully applied to cheese, few studies have described the flavor formation mechanism of the mixed fermentation of *Kluyveromyces marxianus* and *L. paracasei*. The present study aimed to investigate the differences in the flavor compounds of Kazak cheese fermented with single strains (*L. paracasei* SMN-LBK) and mixed strains (*L. paracasei* SMN-LBK and *K. marxianus* SMN-S7-LBK). The LAB and yeast used in this study were isolated from traditional Xinjiang dairy products, which gave the cheese more distinctive Xinjiang characteristics. This study provides evidence regarding the metabolites associated with microbial interaction mechanisms in the mixed fermentation of *L. paracasei* SMN-LBK and *K. marxianus* SMN-S7-LBK.

## 2. Materials and Methods

### 2.1. Fermented Strains and Cheese Fermentation

*L. paracasei* SMN-LBK and *K. marxianus* SMN-S7-LBK, isolated from koumiss, were deposited at the Food Science and Technology Department of Shihezi University. *L. paracasei* SMN-LBK was sub-cultured thrice in MRS medium (Qingdao Hope Bio-Technology Co., Ltd., Qingdao, China) at 37 °C for 14 h, while *K. marxianus* SMN-S7-LBK was sub-cultured thrice in YPD medium (Qingdao Hope Bio-Technology Co., Ltd., Qingdao, China) at 28 °C for 24 h. The cultures were centrifuged (3000× *g*, 15 min) and the cell pellets were washed twice with sterile saline solution. A single-strain inoculum was prepared using *L. paracasei* SMN-LBK (5 × 10^7^ CFU per mL), while a mixed-strain inoculum was prepared using *L. paracasei* SMN-LBK and *K. marxianus* SMN-S7-LBK (5 × 10^7^ CFU per mL per strain). *L. paracasei* SMN-LBK was used as a starter and *K. marxianus* SMN-S7-LBK was used as an adjunct starter to make mixed-fermentation cheese, while *L. paracasei* SMN-LBK was used to make single-strain-fermentation cheese; these were denoted as M and S, respectively.

### 2.2. Cheese Preparation

Standardized bovine milk (2.9% protein and 3.5% fat, by weight) was pasteurized at 63 °C for 30 min. The pasteurized milk was inoculated with 4% (*v*/*v*) freshly prepared single-strain inoculum, stirred, and fermented for 14 h at 37 °C to coagulate. The vessels were sealed with plastic wrap and the fermentation process was carried out under sealed conditions. The curd (pH is about 4.6) was cut into small cubes (1 cm^3^) with a cheese knife when it could be separated from the vessel wall, then filtered through four layers of gauze for 12 h until there was no whey available. The collected curd was divided into two parts; one part was used as S, while the other part was inoculated with 2% (*w*/*w*) adjunct cultures and used for M. Then, the curds were salted (1% salt) and stirred well, put into molds (15 cm × 15 cm × 5 cm), and pressed for 24 h at 300 kPa to further separate the whey. The curds drained of whey were cut into small pieces and vacuum packaged, then stored in an incubator (ShanghaiYiheng Scientific Instrument Co., Ltd., Shanghai, China) at 4 °C at 90% humidity for 40 days. The cheese samples were taken every 10 days, quick-frozen in liquid nitrogen, and then stored at −80 °C until analysis.

### 2.3. Measurement of Microbial Population

The cheese samples were diluted ten times by 0.85% (*w*/*v*) sterile saline and homogenized in an aseptic bag (Qingdao Hope Bio-Technology Co., Ltd., Qingdao, China) for 5 min. LAB in single-strain fermented cheese were counted on MRS agar medium at 37 °C for 36 h. LAB in mixed-fermentation cheese were counted on modified Chalmers medium (soybean peptone 5 g/L, beef powder 5 g/L, yeast extract 5 g/L, glucose 20 g/L, lactose 20 g/L, calcium carbonate 10 g/L, agar 15 g/L, neutral red 0.05 g/L) (Qingdao Hope Bio-Technology Co., Ltd., Qingdao, China) at 37 °C for 36 h, while yeast was counted on YPD agar medium containing chloramphenicol (100 mg/L) at 28 °C for 48 h.

### 2.4. Determination of Physicochemical Properties

The protein content, sodium chloride content, moisture content, and pH of cheese were measured as previously described [30]. The moisture content was calculated using the following formula.
Moisture content of cheese (%) = (wet weights of cheese − dry weights of cheese)/wet weights of cheese × 100.

### 2.5. Determination of Free Amino Acids

The free amino acids (FAAs) in the cheese samples were determined following the method of Buňka et al. [31] in an amino acid analyzer (LBA800; Tianjin Rambo Co., Ltd., Tianjin, China).

### 2.6. Determination of Volatile Compounds

The volatile compounds in cheese samples were analyzed on a gas chromatography–mass spectrometry system (GC-MS; Agilent Technology Co., Ltd., Santa Clara, CA, USA), as previously described [30]. The temperature program conditions were as follows: Starting temperature of 35 °C for 5 min, increased to 100 °C at a rate of 5 °C/min, then increased to 180 °C at a rate of 6 °C/min, before finally increasing to 230 °C at a rate of 8 °C/min. The mass spectrometry system was operated with a heated electrospray ionization source, and the parameters were set as follows: 200 °C ion source temperature, 150 °C quadrupole temperature, and 70 eV electron energy. Volatile compounds were identified via the commercial NIST 2017 database. The volatile compounds were semi-quantified based on their peak area [32].

### 2.7. Determination of Nonvolatile Compounds

Ultra-high performance liquid chromatography (UHPLC; Thermo Fisher Scientific, Shanghai, China) was used to separate nonvolatile compounds in samples as previously described [33,34]. The liquid chromatography conditions were as follows: Column, Waters ACQUITY UPLC BEH Amide (2.1 mm × 100 mm, 1.7 μm); mobile phase, acetonitrile/aqueous solution (25 mmol/L ammonium acetate and 25 mmol/L ammonia); injection volume, 2 μL.

The mass spectrometer (Q Exactive HFX; Thermo Fisher Scientific, Shanghai, China) was used to collect MS/MS spectra under the following conditions: sheath gas flow rate, 30 a.u.; auxiliary gas flow rate, 25 a.u.; capillary temperature, 350 °C; full MS resolution, 60,000; MS/MS resolution, 7500; collision energy, 10/30/60 in NCE mode; spray voltage: 3.6 kV (positive) or −3.2 kV (negative). The substance peak was matched with the in-house MS2 database BiotreeDB for substance annotation [33].

### 2.8. Statistical Analysis

The statistical significance between the index and metabolite was determined using Duncan’s test (*p* < 0.05) with DPS. Significant differences in single-strain-fermentation and mixed-fermentation cheeses were determined by orthogonal projections to latent structures discriminant analysis (OPLS-DA) and PCA using SIMCA 14.1. TBtools v1.068 was also used for the cluster heatmap analysis of flavor compounds.

## 3. Results and Discussion

### 3.1. Microbial Analysis of Cheese

During ripening, starter strains are subjected to various stresses, such as low pH, low temperatures, and substrate availability. Strains that remain active under stresses and have metabolic activity are positive for ripening [35]. The viable LAB count was within a limit of 10^8^–10^9^ cfu/g, and the viable yeast count was within a limit of 10^5^–10^6^ cfu/g (Figure 1). This was consistent with existing reports [36]. The viable LAB count remained stable in single-strain fermentation during the overall low-temperature ripening process. Stimulatory or inhibitory interactions exist between yeast and LAB, and the stability of yeasts and LAB is associated with the strain combination [37]. The variation tendency of LAB was similar to that of yeast in M, which suggests that the two strains form a symbiotic relationship in the later stages of ripening. Similar results were also reported by Liu et al. [38], who noted that yeast and LAB can also form a mutualistic symbiotic relationship in acid rice soup. The symbiotic relationship between yeast and LAB can be explained in two points. Firstly, NH_3_-N increases in cheese during the later stage of ripening [39], and yeast can utilize ammonium to reduce the consumption of AAs [40]. Secondly, some AAs (such as valine or leucine) excreted by yeasts can promote LAB growth [41].

### 3.2. Analysis of Physicochemical Properties of Cheese

The compositions present during cheese ripening are shown in Table 1. No significant differences in the moisture, protein, and salt contents were observed during the ripening of S and M. These results were consistent with a previous study [42], since vacuum polyethylene film packaging limits moisture loss, and protein and salt contents vary in proportion to moisture. The differences in physicochemical properties between S and M were not significant, indicating that the physicochemical properties were not affected by the yeast-assisted starter.

The pH of S decreased first and then was almost unchanged, while the pH of M showed no significant difference. This might be due to the pH being related to the strain interaction and lactose metabolism balance. The fermentation of LAB produces lactic acid, which lowers the pH, while mixed fermentation increases the complexity of the system. Acids are converted into other aromatic components [26]; yeasts can metabolize the lactic acid produced by LAB to CO_2_ and H_2_O [43]; and proteolysis leads to the production of ammonia [44], which might prevent the pH from continuing to decrease.

### 3.3. Analysis of FAA in Cheese

The catabolism of AAs by microorganisms leads to the production of flavor compounds, which is a major process in the development of cheese flavor [45]. The typical taste of cheese can be contributed by AAs and derivatives. A bitter taste is associated with His, Lys, Val, Tyr, Phe, Ile, and Leu; an umami taste is related to Glu, Tyr, and Asp; and a sweet taste is attributed to Met, Ala, Gly, Pro, and Ser [46,47]. The main AAs found were Asp, Glu, Leu, and Lys. The Glu content was the highest and the Cys content was the lowest in the two types of cheese (Figure 2). This result suggested that *K. marxianus* did not affect the AA composition in the process of fermentation and ripening.

The content of most FAAs (except for Lys and His) in M was higher than that in S in the early stage of ripening. This may be due to the fact that mixed fermentation leads to higher levels of proteolysis and AA production. The content of most FAAs (such as Leu and Val) in M was lower than that in S in the later stage of ripening, which was possibly due to the fact that M obtained more aromatic components that had been converted from amino acids. AAs from casein hydrolysis can be converted into precursors for flavor components such as alcohol, aldehydes, ketones, and fatty acids [48]. AAs generate α-keto acids through transamination, α-keto acids are decarboxylated to generate aldehydes, aldehydes are reduced to alcohols, and alcohols also undergo esterification to generate esters [16]. Yeast can produce aromatic compounds associated with the catabolism of branched-chain amino acids, aromatic amino acids, and sulfur-containing amino acids [49]. Leu, Val, and Phe are converted to 3-methyl-butanal, 3-methyl-1-butanol, and benzene ethanol through the Ehrlich pathway [45].

### 3.4. Analysis of Volatile Compounds in Cheese

GC-MS has been used in the comparative metabolomics of different cheese species through untargeted metabolite profiling analysis. A total of 47 volatile flavor compounds were identified in S and M, including 9 alcohols, 8 aldehydes, 11 acids, 5 ketones, and 14 esters (Appendix A). The scattered points of M in the middle and later stages of ripening were clearly distinguished from those of other samples. It can be speculated that the transformation of the substances of M was violent in the middle stage of ripening and that the accumulation of substances was the most abundant in the later stage of ripening (Figure 3A). In the two types of cheese, esters were the most abundant volatile compounds during ripening, followed by acids and alcohols (Figure 3B). The proportions of alcohol, aldehyde, acid, and ketone volatile compounds of M in the middle and later ripening stages were similar to those of S in the later ripening stage, indicating that mixed fermentation promoted the production of volatile compounds. The volatile compounds of S and M were divided into five groups in the heatmap (Figure 3C). The third group of volatile compounds was mainly esters, which were significantly more abundant in M than in S. In general, the types and quantities of ester compounds in M were higher than those in S.

The main alcohols present during the cheese ripening process were 2-methyl-1-propanol, benzene ethanol, and 3-methyl-1-butanol. In the early and middle stages of ripening, the alcohols in M showed a different change trend to those in S. The change trends of the 2-methyl-1-propanol and benzene ethanol contents at 10–30 days and the 3-methyl-1-butanol content at 1–20 days in S were different to those in M. The contents of 2-methyl-1-propanol, benzene ethanol, and 3-methyl-1-butanol in M were higher than those in S, finally reaching 0.086, 3.228, and 4.778 μg/g, respectively. Yeast contributes to the production of higher alcohols and esters with aromatic activity, which have important industrial significance for the aroma contribution of fermented food [50]. The biosynthesis of higher alcohols involves the decarboxylation of α-keto acids to form aldehydes, which are then reduced to corresponding higher alcohols [51]. Val, Leu, Ile, Met, Tyr, Trp, and Phe can be transaminated to α-keto acids [52]. Val, Phe, and Leu are decomposed to form 2-methyl-1-propanol, benzene ethanol, and 3-methyl-1-butanol, which corresponded to the changes in AA content [1]. The high contents of alcohols in S and M might be due to the enzymes produced by microorganisms in these two cheeses having a positive role in alcohol production [53].

The oxidation of lipids, such as unsaturated fatty acids, can produce aldehydes and ketones [54]. The most abundant aldehydes present during the cheese ripening process were hexanal, nonanal, heptanal, and benzaldehyde. A higher hexanal content was detected for M than for S during ripening. Hexanal gives cheese a grassy and rancid flavor, and is an important odor-active carbonyl compound from lipid oxidation [55]. In mature cheese, the aldehyde content is lower than those of ketones and acids, which is a positive change since a high aldehyde content can result in bad flavor [56].

The β-oxidation or autoxidation of fatty acids can produce methyl ketones in cheese [57]. The most abundant ketones during the cheese maturation process were 2-heptanone, 2-nonanone, and 3-hydroxy-2-butanone. The contents of these ketones in cheese first decreased and then increased. The content of 2-heptanone and 3-hydroxy-2-butanone in M was lower than S in the early and middle stages of ripening. 3-Hydroxy-2-butanone, which is easily oxidized to the typical flavor compound diacetyl [58], was the main ketone volatile compound in our cheeses.

The main sources of acid volatile compounds in cheese are lactic acid metabolism, proteolysis, and lipolysis [44]. The main acids present in the cheese ripening process are acetic acid, butanoic acid, hexanoic acid, heptanoic acid, octanoic acid, nonanoic acid, and decanoic acid [59]. In our cheeses, the major characteristic acid volatile compounds included acetic acid, butanoic acid, hexanoic acid, and octanoic acid, which increased in content with ripening. A lower content of acetic acid and butanoic acid was observed in M than in S in the early and middle stages of ripening, which was probably since more ethyl acetate and ethyl butanoate were produced by esterification in M. Acetic acid is described as creating a sour and vinegary odor; butanoic acid provides a rancid and cheesy odor; and hexanoic acid has a sour, rancid, and goat-like odor [60,61]. The acetic acid content of the experimental cheese (1.974 and 2.596 μg/g) was higher than that previously reported (1.902 μg/g), as were the butanoic acid (experimental cheese, 0.943 and 1.057 μg/g) and hexanoic acid contents (experimental cheese, 1.031 and 1.287 μg/g) [62]. 3-methyl-1-butanal, 3-methyl-1-butanol, and 3-methyl-butanoic acid are the secondary metabolites of casein proteolysis [63].

The esters in cheese are produced by the esterification of alcohols and acids and the alcoholysis of glycerides and alcohols [64]. Esters, especially acetic esters, are related to their corresponding alcohol concentration, which is important due to the highly desired fruit, honey, and perfume-like flavor they provide to fermented foods [65]. The main esters produced during the cheese ripening process were ethyl acetate, 3-methyl-1-butyl acetate, ethyl 2-hydroxypropanoate, ethyl octanoate, ethyl decanoate, 2-phenylethyl acetate, ethyl butanoate, and ethyl nonanoate. The main esters had different contents in the single-strain- and mixed-fermentation cheeses over a period of 10–30 days. The ester contents in M were higher than those in S, with ethyl hexanoate, ethyl 2-hydroxypropanoate, ethyl octanoate, and 2-phenylethyl acetate contents of up to 0.951, 0.963, 0.904, and 1.249 μg/g, respectively. The formation of esters in cheese is largely influenced by yeast metabolism [66], and *K. marxianus* is a good producer of esters and alcohols [67]. In our results, we found that *K. marxianus* particularly contributes to the production of ethyl hexanoate, ethyl 2-hydroxypropanoate, ethyl octanoate, and 2-phenylethyl acetate in cheese. Ethyl esters can improve the balance of the overall cheese flavor and are considered to have a fruity flavor, such as apple, banana, and pineapple flavors [68]. The presence of volatile esters in very small amounts affects the flavor of food [69].

### 3.5. Analysis of Differential Nonvolatile Compounds in Cheese

PCA is a statistical approach used to achieve dimension reduction from a series of possibly related observed variables to linear unrelated variables through orthogonal transformation to extract features and determine the relationship between variables [70]. The nonvolatile compounds produced were significantly different according to the fermentation method used in the ripening process, and changes in metabolites mainly occurred in the middle and later stages of ripening, indicating that mixed fermentation was important for the formation of nonvolatile compounds (Figure 4).

Further analyses and a pairwise comparison of all conditions—namely, M1/S1, M20/S20, and M40/S40 (see Figure 5A)—were conducted. The contribution rates of the principal components of different groups reached 47.2%, 56.6%, and 61.2%, respectively. The mixed and single-strain fermentations were separated into different groups, which improved the clustering effect of the biological replicate samples. No overlap was observed between every two groups, which indicated that mixed fermentation had special variables. The original model explained the difference between the two sets of samples well (R^2^Y and Q^2^ were close to 1) (Figure 5B). OPLS-DA classified the cheese samples based on metabolite and reflected on the effect of ripening time and cheese type on compound variation.

In the volcano map, each scatter point represents a differential metabolite (see Figure 6). The analysis groups showed significantly different distributions and aggregations of scatter points. According to the secondary mass spectrometry database, a total of 397 differentially expressed metabolites were identified between S and M during ripening. In ESI+ ionization mode, the identified nonvolatile compounds were selected, including 143 lipids and lipid-like molecules, 82 organic acids and derivatives, 54 organic heterocyclic compounds, 43 organic oxygen compounds, 17 organic nitrogen compounds, and 17 benzenoids. Among the nonvolatile metabolites in M1, 94 were upregulated and 85 were downregulated compared with S1 (*p* < 0.05); among nonvolatile metabolites in M20, 61 were upregulated and 207 were downregulated compared with S20; among nonvolatile metabolites in M40, 70 were upregulated and 111 were downregulated compared with S40. Lipids and lipid-like molecules (glycerophospholipids and fatty acyls) and organic acids and derivatives (amino acids, peptides, and analogues) were mostly present in the analysis groups. Our results showed that primarily dipeptides and several amino acids were observed as differential sensory nonvolatile metabolites between M and S, mainly comprising Pro, Ala, Gln, Leu, and Ile-related compounds (Appendix A).

### 3.6. Enrichment Analysis of Differential Metabolites

According to the KEGG Pathway database, identified nonvolatile compounds were mapped and annotated onto metabolic pathways. A total of 31 pathways (primarily AA metabolism) were involved in the biosynthesis and catabolism of differential metabolites, such as D-glutamine and D-glutamate metabolism; alanine, aspartate, and glutamate metabolism; arginine and proline metabolism; phenylalanine metabolism; D-alanine metabolism; cysteine and methionine metabolism; and tyrosine metabolism. The analysis of metabolic pathways is shown in a bubble chart in Figure 7. The pathway impact showed that the strain difference caused changes in alanine, aspartate, and glutamate metabolism; D-glutamine and D-glutamate metabolism; and nicotinate and nicotinamide metabolism, while the ripening time caused changes in glycerophospholipid metabolism. Many AAs were identified in our large-scale metabolite analysis and KEGG enrichment analysis. Among the pathways that caused differences in the metabolic profiles, most were related to AAs. Therefore, AA metabolism might be the main cause of differences in sensory characteristics between single-strain and mixed-strain fermentations.

### 3.7. Metabolic Pathway Analysis of Differential Metabolites

Glu and Gln were the most abundant AAs in M. The results obtained for the targeted and untargeted metabolomics were consistent, confirming AA exchanges as potential biomarkers and, in particular, the cross-utilization of Glu and Gln (Figure 8). Therefore, this study focused on analyzing D-glutamine and D-glutamate metabolism, nitrogen metabolism, and phenylalanine metabolism.

In cheese, nitrogen metabolism is related to many aroma compounds, with the precursors of aroma compounds generated by casein hydrolysis [63]. In LAB, milk proteins are degraded into peptides by cell-envelope proteases (CEP) and then further degraded into smaller peptides and amino acids by transport systems and peptidases [71]. In Prato cheeses, the His, Tyr, and Phe contents were increased using arginine and yeast extract [72]. Amino acids can undergo decarboxylation or deamination to form volatile compounds (amines, carboxylic acids, and ammonia) and can be converted into α-keto acids and then α-ketoglutarate, finally producing glutamic acid [16].

Glu (glutamate) is a representative substance with an umami taste; it can improve positively perceived flavors (sweet taste and salty taste) and suppress less acceptable flavors (sourness and bitterness) [73]. Gln and Glu can increase the acid resistance and survival of LAB since intracellular decarboxylation consumes intracellular protons [74]. In bacteria, glutamine-amidotransferases or glutaminases catalyze the conversion of Gln to Glu. Gln deamination promotes the production of potential functional metabolites (such as γ-aminobutyrate) [75], and the acid resistance of *Lacticaseibacillus* is enhanced by the electrogenic antiport of Glu and GABA in Glu decarboxylation [76]. Glu or Gln conversion contribute to the survival of *Limosilactobacillus reuteri* under acidic conditions similar to the acidity of experimental cheese (around pH 4.0) [77].

Phe is catabolized by *Lacticaseibacillus casei* strains through successive, constitutively expressed transamination and dehydrogenation reactions. Phe catabolism mainly produces phenyllactic acid, phenylacetic acid, and benzoic acid. The spontaneous chemical degradation of a Phe intermediate, phenylpyruvic acid, can produce phenylacetic acid, benzoic acid, benzeneethanol, phenylpropionic acid, and benzaldehyde [78].

Despite the advantages of the use of single strains in controlling the fermentation process, some consumers think that single-strain fermentation lacks flavor complexity compared with mixed fermentation [79]. During the mixed fermentation of yeast with LAB, different interactions between strains have effects, especially competition for nitrogen, with the nitrogen requirements and order of assimilation of nitrogen substrates being different among different strains [80]. As starter cultures, the adaptation of strains to specific habitats is closely associated with their functionality owing to the presence of metabolic activity in cheese.

## 4. Conclusions

This study represents complete information regarding the changes in metabolites between single-strain-fermentation cheese and mixed-fermentation cheese during ripening. In total, 47 volatile and 397 nonvolatile compounds were identified as the main differential compounds in single-strain-fermentation cheese and mixed-fermentation cheese. Acids, alcohols, and esters were the main volatile flavor compounds in mixed-fermentation cheeses, of which ethyl esters had the highest content and contributed to the production of floral and fruity aromas. Pro, Ala, Gln, Leu, Ile, and related dipeptides were the main differential nonvolatile flavor compounds and were involved in the formation of the unique taste of cheese. The metabolic pathways of glutamate and glutamine had a higher correlation with the differences in nonvolatile metabolites. Metabolic pathway analysis showed that a number of potentially functional flavor compounds were involved in AA metabolism. This research provides interesting perspectives for the production of mixed-fermentation cheeses with flavor enhancement and will help with the identification of flavor formation mechanisms and microbial interactions.

## Figures and Tables

**Figure 1 foods-11-00966-f001:**
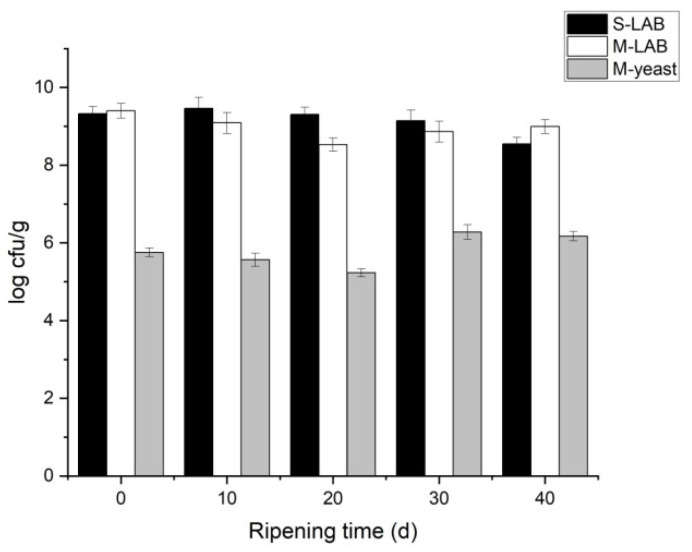
LAB and yeast content of cheese (log cfu/g). Abbreviations: M, mixed-fermentation cheese; S, single-strain-fermentation cheese.

**Figure 2 foods-11-00966-f002:**
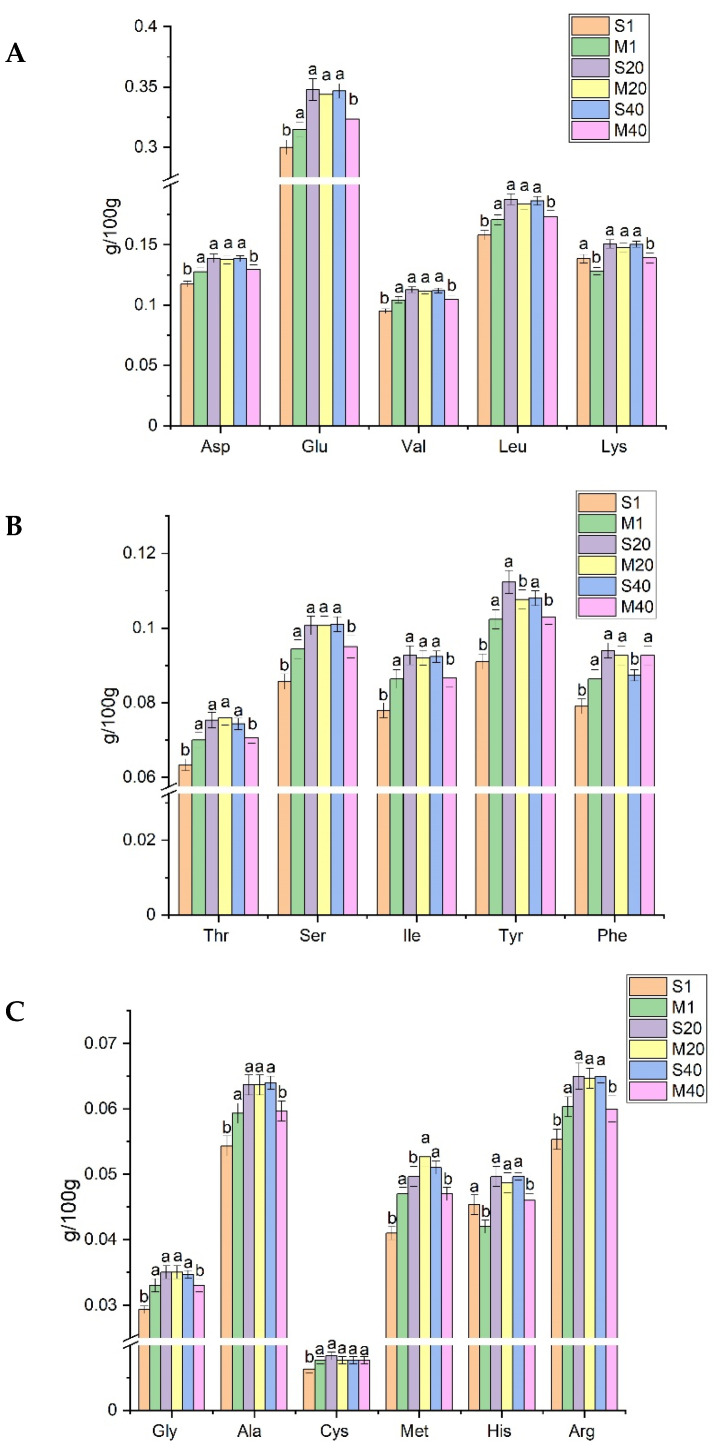
FAA contents of cheese (g/100 g). (**A**–**C**) are the content of different kinds of FAAs. Different letters indicate a significant difference between mixed-fermentation and single-strain-fermentation cheese obtained on the same day (*p* < 0.05). Abbreviations: M, mixed-fermentation cheese; S, single-strain-fermentation cheese. Numbers 1–40 represent different ripening times (in days).

**Figure 3 foods-11-00966-f003:**
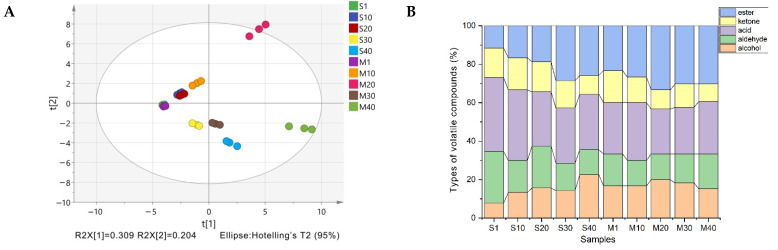
(**A**) Principal component analysis of cheese samples based on volatile compounds. (**B**) Types of volatile compounds in cheese. (**C**) Hierarchical cluster analysis of cheese volatile compounds during ripening. Individual cells corresponded to different cheeses and variables, which are colored red and blue to indicate high and low abundances of these variables, respectively. Abbreviations: M, mixed-fermentation cheese; S, single-strain-fermentation cheese. Numbers 1–40 represent different ripening times (in days).

**Figure 4 foods-11-00966-f004:**
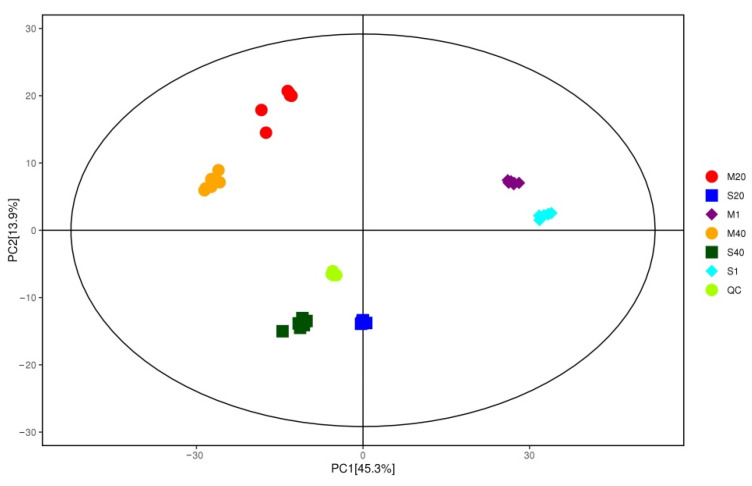
Score scatter plot of PCA model for all groups. Abbreviations: M, mixed-fermentation cheese; S, single-strain-fermentation cheese. Numbers 1–40 represent different ripening times (in days).

**Figure 5 foods-11-00966-f005:**
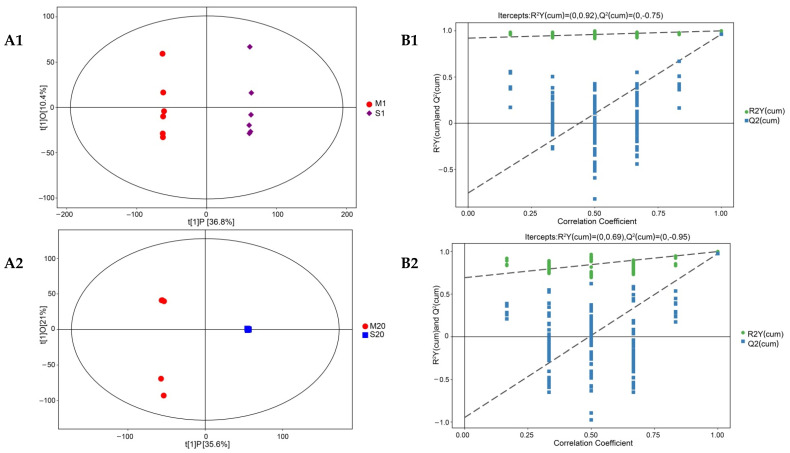
Score scatter plot and permutation test of OPLS-DA model for different groups. (**A1**–**A3**) represent score scatter plots of the OPLS-DA model of M1 vs. S1, M20 vs. S20, and M40 vs. S40. (**B1**–**B3**) represent permutation tests of the OPLS-DA model of M1 vs. S1, M20 vs. S20, and M40 vs. S40. The original model R^2^Y was very close to 1, indicating that the established model conformed to the real situation of the sample data. The original model, Q^2^, was very close to 1, indicating that if a new sample was added to the model, an approximate distribution would be obtained. Abbreviations: M, mixed-fermentation cheese; S, single-strain-fermentation cheese. Numbers 1–40 represent different ripening times (in days).

**Figure 6 foods-11-00966-f006:**
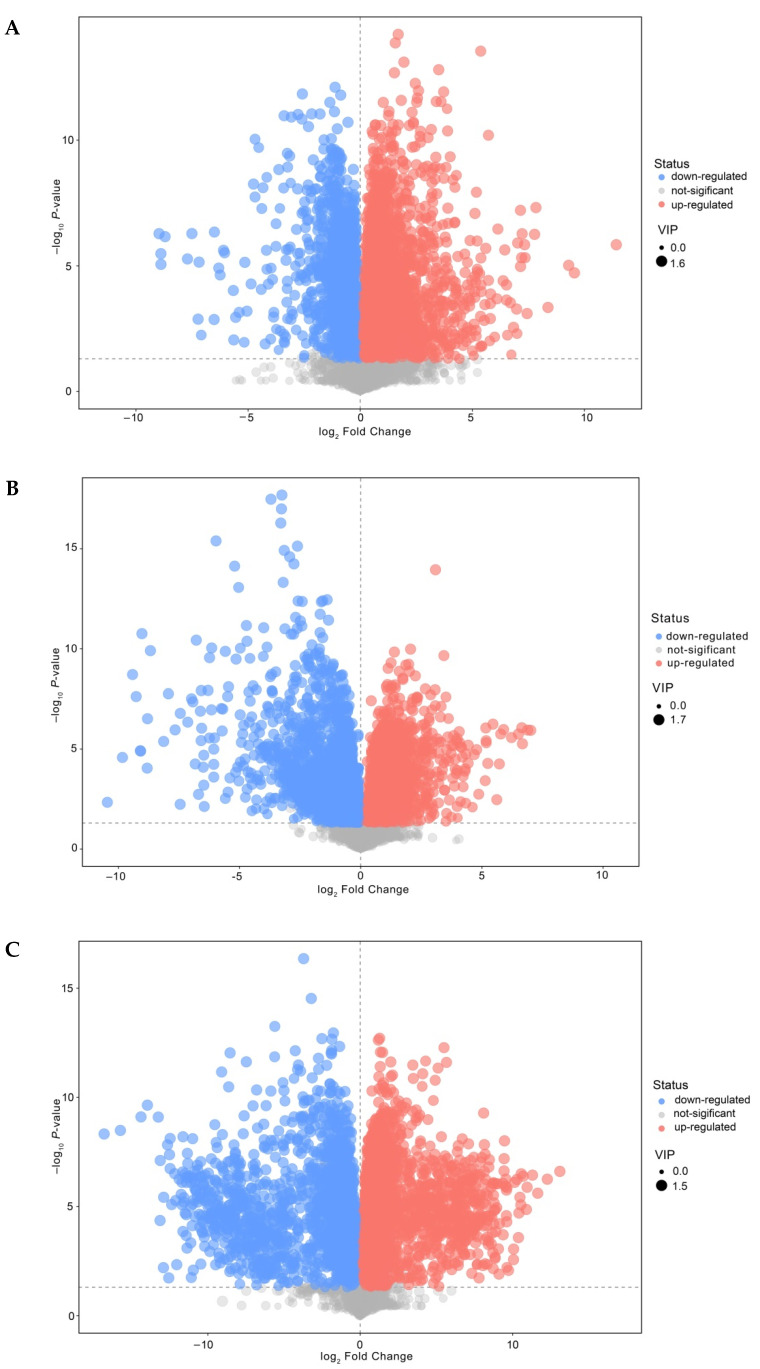
Volcano plot of OPLS-DA model for different groups. (**A**–**C**) are volcano plots of M1 vs. S1, M20 vs. S20, and M40 vs. S40. The size of the points represents the VIP value according to OPLS-DA. The abscissa represents the multiple change of the group compared to each substance, and the ordinate represents the *p*-value of the Student’s *t*-test. The scattered colors represent the final screening results. Metabolites that are significantly upregulated are shown in red, metabolites that are significantly downregulated are shown in blue, and metabolites that are not significantly different are shown in gray. Abbreviations: M, mixed-fermentation cheese; S, single-strain-fermentation cheese. Numbers 1–40 represent different ripening times (in days).

**Figure 7 foods-11-00966-f007:**
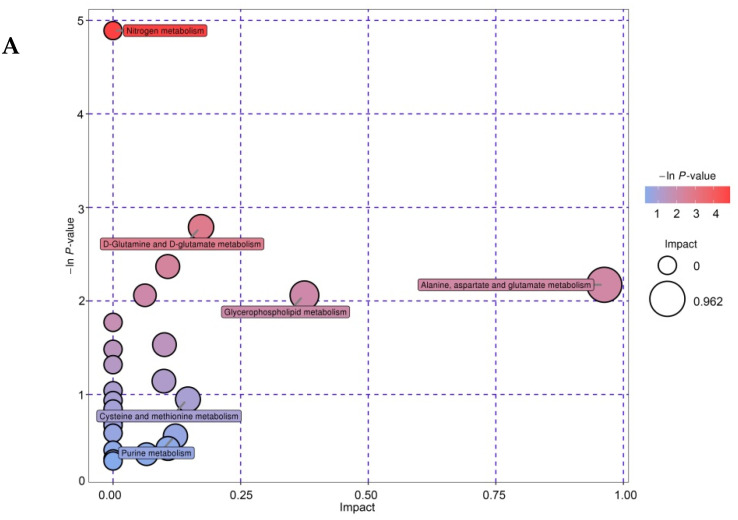
Enrichment analysis of differential metabolites. Each bubble in the bubble chart represents a metabolic pathway. The abscissa and size of the bubble indicate the influence of the factor on the pathway in the topology analysis. The p value of the enrichment analysis is presented as a bubble color in the bubble ordinate. A darker color results in a smaller p value and a more significant degree of enrichment. (**A**–**C**) are bubble plots of M1 vs. S1, M20 vs. S20, and M40 vs. S40. Abbreviations: M, mixed-fermentation cheese; S, single-strain-fermentation cheese. Numbers 1–40 represent different ripening times (in days).

**Figure 8 foods-11-00966-f008:**
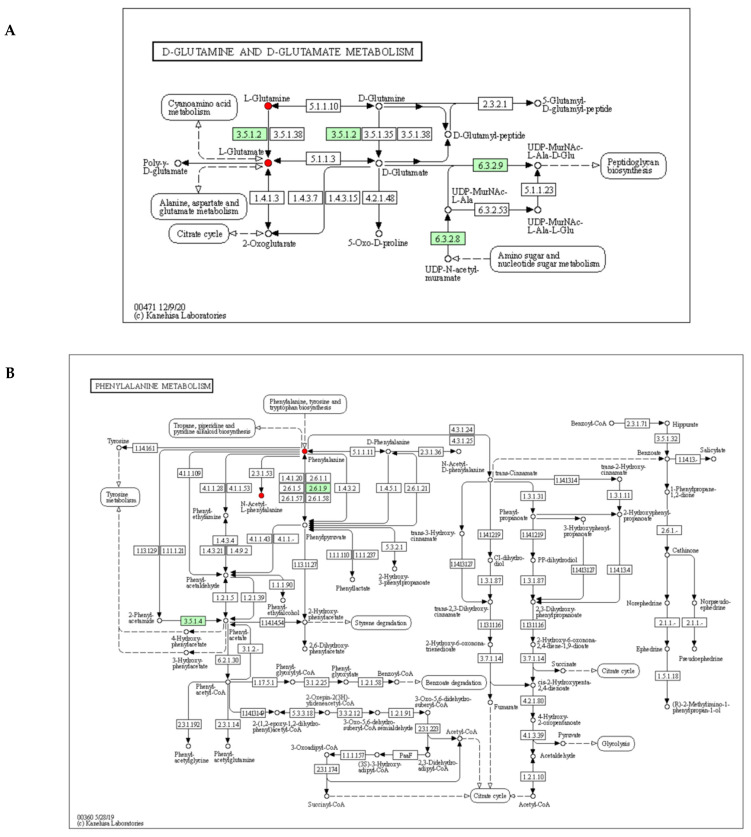
KEGG pathway analysis of differential metabolites. (**A**) D-glutamine and D-glutamate metabolism pathway of M1 vs. S1, (**B**) phenylalanine metabolism pathway of M40 vs. S40. Abbreviations: M, mixed-fermentation cheese; S, single-strain-fermentation cheese. Numbers 1–40 represent different ripening times (in days).

**Table 1 foods-11-00966-t001:** The pH, moisture, protein and salt content of cheese.

	Moisture (%)	pH	Protein (%)	Salt (%)
S1	51.24 ± 0.57 ^Aa^	4.02 ± 0.02 ^Aa^	18.43 ± 0.18 ^Aa^	0.76 ± 0.02 ^Aa^
S10	51.76 ± 0.33 ^Aa^	3.92 ± 0.01 ^Ab^	18.47 ± 0.17 ^Aa^	0.79 ± 0.03 ^Aa^
S20	51.5 ± 0.51 ^Aa^	3.93 ± 0.02 ^Ab^	18.70 ± 0.26 ^Aa^	0.81 ± 0.02 ^Aa^
S30	50.75 ± 0.48 ^Aa^	3.93 ± 0.01 ^Ab^	18.17 ± 0.28 ^Aa^	0.82 ± 0.02 ^Aa^
S40	50.25 ± 0.38 ^Aa^	3.92 ± 0.01 ^Ab^	18.79 ± 0.16 ^Aa^	0.83 ± 0.02 ^Aa^
M1	51.26 ± 0.33 ^Aa^	3.94 ± 0.01 ^Ba^	17.74 ± 0.15 ^Aa^	0.75 ± 0.03 ^Aa^
M10	51.06 ± 0.63 ^Aa^	3.90 ± 0.02 ^Aa^	18.13 ± 0.16 ^Aa^	0.77 ± 0.03 ^Aa^
M20	50.5 ± 0.52 ^Aa^	3.94 ± 0.02 ^Aa^	18.04 ± 0.19 ^Aa^	0.80 ± 0.02 ^Aa^
M30	50.74 ± 0.36 ^Aa^	3.92 ± 0.01 ^Aa^	18.1 ± 0.15 ^Aa^	0.79 ± 0.02 ^Aa^
M40	50.37 ± 0.44 ^Aa^	3.94 ± 0.02 ^Aa^	18.38 ± 0.14 ^Aa^	0.81 ± 0.03 ^Aa^

Abbreviations: M, mixed-fermentation cheese; S, single-strain-fermentation cheese. Numbers 1–40 represent different ripening times (in days). Different capital letters in each column indicate a significant difference between the two cheeses obtained on the same day; different lowercase letters in each column indicate a significant difference between the same cheese obtained on different days (*p* < 0.05).

## Data Availability

Not applicable.

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
