# Peer review of "Differences between Kazak Cheeses Fermented by Single and Mixed Strains Using Untargeted Metabolomics"

_foods, 2022, doi:10.3390/foods11070966_

Round 1

Reviewer 1 Report

The topic of the article falls within the thematic scope of the journal FOODS.

In this study, the possibilities offered by metabolomics were used to assess the effect of mixed fermentation (Kluyveromyces marxianus SMN-S11-LBK and Lacticaseibacillus paracasei SMN-LBK) and single-strain fermentation (L. paracasei SMN-LBK only) on the profiles of volatile and non-volatile compounds in Kazak cheese. The results of these studies clearly showed the difference between single-strain and mixed fermentation cheeses.

I have no comments about the methods used, and about the general presentation of the results and their discussion. But minor remarks, which, however, usually happen - I have quoted below.

Here are some of the minor considerations:

Chapter 1 (Introduction) - I propose to introduce a brief description - characteristics - of this cheese,

Chapter 2.1 and 2.3 – provide the names of producers, city, country

Chapter 3.3. - it seems to me that the authors should slightly reformat the description of the results and adjust their graphical representation (lines 213 - 216).

Chapter 4 - Conclusions - withdraw from the use of the abbreviations M and S.

All comments and suggestions for corrections (including those not listed above) were introduced in the review mode to the attached pdf file.

Author Response

We are very grateful to the reviewers for their hard work and valuable comments regarding our manuscript. We apologize for the impact of our negligence on the reviewer process. We have now revised our manuscript according to the reviewer comments.

Answer to reviewers:

Reviewer #1:

  1. Chapter 1 (Introduction) - I propose to introduce a brief description - characteristics - of this cheese.

Answers: Thank you for this comment. We have added a brief description - characteristics - of this cheese in line 35-36 (page 1).

  1. Line 42: L. paracasei.

Answers: Thank you for this comment. We have changed strain name to its full name when it first appeared in line 45 (page 1). We revised the same question in line 85 (page 2).

  1. Chapter 2.1 and 2.3 – provide the names of producers, city, country.

Answers: Thank you for this comment. We have provided the names of producers, city, country in line 97-100 and line 128 (page 3).

  1. Chapter 3.3. - it seems to me that the authors should slightly reformat the description of the results and adjust their graphical representation (lines 213 - 216).

Answers: Thank you for this comment. We have improved the description of the results in line 225-228 (page 7) and adjusted their graphical representation (page 6-7).

  1. Line 288: Y5 Kazak cheese.

Answers: Thank you for this comment. We have rewritten sentences with unclear statements in line 299-302 (page 9).

  1. Chapter 4 - Conclusions - withdraw from the use of the abbreviations M and S.

Answers: Thank you for this comment. We have withdrawn from the use of the abbreviations M and S in the Conclusions (line 446 of page 16).

  1. Line 514: The information in the references.

Answers: Thank you for this comment. The information in the references should be correct and complete. We have checked and revised the information in the references (line 533 of page 18). We revised the same question in other references.

Reviewer 2 Report

In this manuscript untargeted metabolomics was used to evaluate the impact of mixed fermentation and single-strain fermentation on the volatile and non-volatile compound profiles of Kazak cheese. However, I can some observations to improve the manuscript:

Keywords: Remember that the words should not be the same as those that appear in the title.

Pag. 2, line 92-93. Consider using the word "inoculum" not "inocula".

Results

In figure 3, 5, 6 and 7. I recommend that the font size in the images be uniform. There are some letters that are very small.

In general, it is a good manuscript, it is very well discussed, I congratulate the entire work team.

Author Response

We are very grateful to the reviewers for their hard work and valuable comments regarding our manuscript. We apologize for the impact of our negligence on the reviewer process. We have now revised our manuscript according to the reviewer comments.

Answer to reviewers:

Reviewer #2:

  1. Keywords: Remember that the words should not be the same as those that appear in the title.

Answers: Thank you for this comment. We have changed Keywords in line 29-30 (page 1).

  1. Pag. 2, line 92-93. Consider using the word "inoculum" not "inocula".

Answers: Thank you for this comment. We have replaced "inocula" with "inoculum" in line 101-102 (page 3). We revised the same question in line 111 (page 3).

  1. Results: In figure 3, 5, 6 and 7. I recommend that the font size in the images be uniform. There are some letters that are very small.

Answers: Thank you for this comment. We have uniformed the font size in the images.

Thank you very much for your valuable suggestions and questions, we will continue to work hard.

Reviewer 3 Report

The article reports an evaluation of the impact of mixed fermentation and single-strain fermentation on the volatile and non-volatile compound profiles of Kazak cheese.

The paper can be of good interest for the reader of the journal and the methodological approach appears as consistent, as well as the presented processes.

In my opinion, the paper can be accepted for publication after some changes, according to the following indications:

• In the introduction, the work is poor about the fermentator utilize to the scope. The aim of the work and the novelty with respect to the existing literature should be adequately discussed. To this point, I suggest to enlarge the state-of-the-art analysis by including new parts about utilized fermentation system.

• Moreover, in the final part of the introduction, I suggest to report the content of the paper.

• In materials and methods, there isn’t any description about fermentation system. How do you control the process? In particular, from the plant engineering point of view, how is it managed? How is the homogeneity of the process checked?

• The conclusions are very synthetic too. I think that they should be re-written by indicating the main conclusions achieved in this study.

Author Response

We are very grateful to the reviewers for their hard work and valuable comments regarding our manuscript. We apologize for the impact of our negligence on the reviewer process. We have now revised our manuscript according to the reviewer comments.

Answer to reviewers:

Reviewer #3:

  1. In the introduction, the work is poor about the fermentator utilize to the scope. The aim of the work and the novelty with respect to the existing literature should be adequately discussed. To this point, I suggest to enlarge the state-of-the-art analysis by including new parts about utilized fermentation system.

Answers: Thank you for this comment. We have discussed the aim of the work and enlarge the state-of-the-art analysis by including new parts about utilized fermentation system in line 64-70 and line 83-85 (page 2).

  1. Moreover, in the final part of the introduction, I suggest to report the content of the paper.

Answers: Thank you for this comment. We have reported the content of the paper in the final part of the introduction in line 85-92 (page 2).

  1. In materials and methods, there isn’t any description about fermentation system. How do you control the process? In particular, from the plant engineering point of view, how is it managed? How is the homogeneity of the process checked?

Answers: Thank you for this comment. We have added description about fermentation system in line 115-117 (page 3). We strictly control the fermentation process through temperature, time, sealing conditions. The process control through process parameters from the plant engineering point of view. We have added the conditions of cutting curd, filtering whey, press molding to ensure the homogeneity of the process in line 111-118 (page 3).

  1. The conclusions are very synthetic too. I think that they should be re-written by indicating the main conclusions achieved in this study.

Answers: Thank you for this comment. We have re-written by indicating the main conclusions achieved in this study in line 443-450 (page 16).
